# Derivatization of Natural Compound β-Pinene Enhances Its In Vitro Antifungal Activity against Plant Pathogens

**DOI:** 10.3390/molecules24173144

**Published:** 2019-08-29

**Authors:** Yunfei Shi, Hongyan Si, Peng Wang, Shangxing Chen, Shibin Shang, Zhanqian Song, Zongde Wang, Shengliang Liao

**Affiliations:** 1College of Forestry, Jiangxi Agricultural University; National Forestry and Grassland Bureau Woody Spice (East China) Engineering Technology Research Center; National Forestry and Grassland Bureau/Jiangxi Provincial Camphor Engineering Technology Research Center; Collaborative Innovation Center of Jiangxi Typical Trees Cultivation and Utilization, Nanchang 330045, China; 2Institute of Chemical Industry of Forest Products, Chinese Academy of Forestry, Nanjing 210042, China

**Keywords:** pinene, myrtanyl acid, amide, acylthiourea, antifungal activity

## Abstract

Background: The development of new antifungal agents has always been a hot research topic in pesticide development. In this study, a series of derivatives of natural compound β-pinene were prepared, and the antifungal activities of these derivatives were evaluated. The purpose of this work is to develop some novel molecules as promising new fungicides. Methods: Through a variety of chemical reactions, β-pinene was transformed into a series of β-pinene-based derivatives containing amide moieties and acylthiourea moieties. The antifungal activities of these derivatives against five plant pathogens including *Colletotrichum gloeosporioides*, *Fusarium proliferatum*, *Alternaria kikuchiana*, *Phomopsis* sp. and *Phytophthora capsici* were tested; preliminary structure–activity relationship was discussed. Results: Some derivatives exhibited moderate or significant antifungal activity due to the fusion of the amide moiety or the acylthiourea moiety with the pinane skeleton. The structure–activity relationship analysis showed that the fluorine atom and the strong electron withdrawing nitro group, or trifluoromethyl group on the benzene ring of the derivatives had a significant effect on the improvement of the antifungal activity against *Colletotrichum gloeosporioides*, *Fusarium proliferatum*, *Alternaria kikuchiana* and *Phomopsis* sp. Meanwhile, the introduction of an ethyl group at the meta-position on the benzene ring of the derivatives could improve the antifungal activity against *Phytophthora capsici*. Compounds **4e**, **4h**, **4q**, **4r** exhibited broad-spectrum antifungal activity against the tested strains. Compound **4o** had significant antifungal activity against *Phytophthora capsici* (IC_50_ = 0.18 μmol/L). These derivatives were expected to be used as precursor molecules for novel pesticide development in further research.

## 1. Introduction

During the growth of crops, various plant diseases occur; 70% to 80% of plant diseases are caused by phytopathogenic fungi infection [1,2]. When infecting crops, plant pathogenic fungi can not only directly reduce the yield and quality of crops, but also secrete a variety of toxins and metabolites that are harmful to humans and animals, causing a huge threat to human health [3]. Therefore, effective and safe inhibition of growth of plant pathogenic fungi has become the key to control plant diseases.

Chemical antifungal agents can effectively kill or inhibit the growth of plant pathogenic fungi, and play a vital role in protecting human health, increasing crop yields, and food preservation [4,5]. However, a long-term usage of the same antifungal agent often leads to an increase in the resistance of phytopathogenic fungi [6,7]. Continuous development of new antifungal agents is still the active demand, and the development of new antifungal agents has also been a hot spot in pesticide research.

The compound β-pinene is a natural compound with antimicrobial activity [8,9,10] which can be used to participate in numerous chemical reactions. Masses of β-pinene derivatives can be synthesized by chemical modification, and some of these derivatives have been proved to have better antifungal activity [11,12,13,14,15,16,17]. For example, Li et al. [14] synthesized three series of β-pinene-based derivatives, and tested their fungicidal activity against three agricultural fungi. As a result, there two derivatives displayed potent fungicidal activity against *Rhizoctonia solani*, with IC_50_ values of 2.439 and 1.857 mg/mL. Thus, β-pinene was treated as a good precursor compound for the development of plant-derived pesticides. Based on previous results, in this study, β-pinene was used as a raw material to design and synthesize a series of derivatives through splicing of different bioactive substructures, aiming to obtain derivatives with stronger antifungal activity against plant pathogenic fungi. This study can provide a reference for the development of novel potent plant-derived antifungal agents.

## 2. Results

### 2.1. Chemisty

We carried out a series of modifications of β-pinene (Figure 1). Firstly, β-pinene was converted to myrtanol by a hydroboration reaction, and the myrtanol was oxidized to myrtanyl acid by an oxidation reaction. Then, based on the substructure splicing principle, a series of amide derivatives and acylthiourea derivatives were synthesized by splicing an amide moiety or an acylthiourea moiety onto the pinane skeleton.

The molecular structures of β-pinene-based derivatives were characterized by IR, NMR and MS. The results showed that the target derivatives **2**, **3**, **4a**–**4l** and **4m**–**4u** were successfully synthesized in this study.

Taking compound **4k** as an example to analyze the characterization of β-pinene-based amide derivatives, the FT-IR spectrum of **4k** showed an absorption band at 3432 cm^−1^ due to the new amide N–H group and the characteristic stretching frequency of C=O was observed at 1687 cm^−1^ and 1654 cm^−1^. In the ^1^H NHR spectrum of compound **4k**, the proton signal of N–H was observed as a singlet at δ 11.91 ppm. Aliphatic protons were observed in the range of δ 0.85–3.24 ppm. Aromatic protons appeared in the range of δ 7.02–7.64 ppm. The signal of the thiazole proton appeared in δ 7.74 ppm. In the ^13^C NMR spectrum of compound **4k**, the characteristic amide C=O was observed at 173.66 ppm which confirmed the amide bond was formed between the pinane skeleton and phenylthiazole. The mass spectrum of **4k** showed an expected molecular ion peak at m/z 345.1 [M + H] ^+^, 343.1 [M − H]^−^ corresponding to molecular formula.

In addition, taking compound **4m** as an example to analyze the characterization of β-pinene-based acylthiourea derivatives, the FT-IR spectrum of **4m** exhibited an absorption wavelength at 1686 cm^−1^ corresponding to a C=O group, and the absorption wavelengthsat 1143 cm^−1^ and 1102 cm^−1^ belonging to a C=S group. The acylthiourea N–H group was observed at 3635 cm^−1^. The characteristic absorption band of the C–N group of acylthiourea appeared at 1327 cm^−1^, 1310 cm^−1^ and 1296 cm^−1^. In the ^1^H NHR spectrum of compound **4m**, the signal related to the proton of thiourea at δ 12.47 ppm appeared. All other aromatic and aliphatic region protons were observed at an acceptable range. In the ^13^C NMR spectrum of compound **4m,** the C=O and C=S groups of the acylthiourea were observed in δ 178.33 ppm and δ 176.81 ppm, respectively. The mass spectrum of **4m** showed an expected molecular ion peak at *m*/*z* 325.1 [M + Na] ^+^ and 301.2 [M − H]^−^ corresponding to the molecular formula.

The spectral data of these compounds can be found in the Appendix A.

### 2.2. Biological Activity

The antifungal activities of the β-pinene-based derivatives against five plant pathogens are shown in Table 1. The antifungal activity of the β-pinene-based derivatives against *Colletotrichum gloeosporioides* showed that there were five derivatives (**4e**, **4g**, **4h**, **4j** and **4r**) with IC_50_ values less than 200 μmol/L. Among them, the acylthiourea derivative **4r** had the highest antifungal activity, and its IC_50_ value was 21.64 μmol/L. For the *Fusarium proliferatum*, it was shown that there were five compounds (**4a**, **4e**, **4g**, **4h** and **4q**) with IC_50_ values below 200 μmol/L, and the amide derivatives **4e** and **4h** had better activities; their IC_50_ values were 39.21 μmol/L and 41.98 μmol/L, respectively. For the *Alternaria kikuchiana*, the IC_50_ values of five amide derivatives (**4a**, **4b**, **4c**, **4e** and **4h**) were less than 200 μmol/L, and compound **4e** exerted the best antifungal activity (IC_50_ = 38.8 μmol/L). The antifungal activity of the β-pinene-based derivatives against *Phomopsis* sp. showed that most of the derivatives had certain inhibitory effects on *Phomopsis* sp., and the IC_50_ of 11 derivatives (**4a**, **4c**, **4d**, **4e**, **4h**, **4i**, **4j**, **4m**, **4o**, **4q** and **4r)** were less than 200 μmol/L. The amide derivative **4h** had the best effect on the *Phomopsis* sp., and its IC_50_ value was 20.43 μmol/L.

Compared with the intermediate compound myrtanyl acid, only compounds **4d**, **4o**, **4q** and **4r** exhibited better antifungal activity against *Phytophthora capsici*. Among them, compound **4o** displayed potent antifungal activity against *Phytophthora capsici*, and its IC_50_ value was 0.18 μmol/L, which was lower than that of the positive control carbendazim against *Phytophthora capsici*.

## 3. Discussion

Both the amide structure and the acylthiourea structure are important antifungal active sub-structural units. Amide compounds have been used as fungicides for decades. Up to now, more than 30 varieties have been commercialized, such as ethaboxam, metalaxyl, thifluzamide, etc. (see Figure 2).

Acylthioureas also exhibit good antifungal activity. For example, Antypenko et al. reported a series of acylthiourea derivatives with good activity against fungi and *Phytophthora* pathogens [18]. Qin et al. reported that chitosan-based thiourea derivatives containing 1,2,4-triazole heterocycles had good antifungal activity against plant pathogenic bacteria [19]. Li et al. reported that a series of decanoic acid thiourea derivatives have good antifungal activity against various plant pathogenic fungi [14]. Gao et al. reported that a series of terpene-based acyl-thiourea derivatives showed moderate to excellent antifungal activities against several fungi [16,17].

Since the amide structure and the acylthiourea structure are the key structures for exerting antifungal activity, when designing the β-pinene-based derivatives, we blended the amide moiety or the acylthiourea moiety into the pinane skeleton, in order to obtain new hybrids with good antifungal activity.

The antifungal activity of the synthesized derivatives against five plant pathogens was tested in this work. These five plant pathogens were *Colletotrichum gloeosporioides*, *Fusarium proliferatum, Alternaria kikuchiana*, *Phomopsis* sp. and *Phytophthora capsici*. *Colletotrichum gloeosporioides* is one of the most serious pathogenic fungi of *Camellia oleifera*. If infected with this fungus, the yield of *Camellia oleifera* will be reduced significantly [20]. *Fusarium proliferatum* is a cause of many crop diseases, including rice spikelet rot disease [21], stem rot of *Hylocereus polyrhizus* [22], root rot in soybean [23], etc. *Alternaria kikuchiana* is the pathogen of pear black spot disease. It infects the fruits, leaves and shoots of different varieties of pear trees, causing the leaves of pear trees to fall, early fruit drop, fruit decay and shedding, post-harvest decay, etc [24]. *Phomopsis* sp. may cause leaf spot, fruit rot and postharvest fruit rot of kiwifruit [25,26]. *Phytophthora capsici* can infect various growth stages of pepper plants, causing seedling death, root rot, crown rot, stem blight, leaf spot and fruit rot [27,28]. Hence, it is of great significance to develop new antifungal agents for controlling these fungi.

The results of antifungal activity test showed that compared with the raw material β-pinene and the intermediate compound myrtanyl acid, β-pinene-based derivatives that were synthesized by the blend of amide group and pinane skeleton or the blend of acylthiourea group and pinane skeleton exerted better antifungal activity against plant pathogens. Some derivatives exhibited moderate to significant antifungal activity. In general, the antifungal activity of amide derivatives against *Colletotrichum gloeosporioides*, *Fusarium proliferatum* and *Alternaria kikuchiana* was better than that of acylthiourea derivatives. However, the antifungal activity of acylthiourea derivatives against *Phytophthora capsici* was better than that of amide derivatives.

Although it is hard to give a comprehensive structure–activity relationship to these derivatives, we can find some interesting hints from Table 1. For *Colletotrichum gloeosporioides*, the four most potent compounds were **4r** < **4h** < **4e** < **4g**. For *Fusarium proliferatum*, the three most potent compounds were **4e** < **4h** < **4g**. For *Alternaria kikuchiana*, the three most potent compounds were **4e** < **4h** < **4c**. For *Phomopsis* sp., the three most potent compounds were **4h** < **4e** < **4q**. These results revealed that an introduced fluorine atom (**4r** and **4g**), nitro group (**4h**) or trifluoromethyl group (**4e**) at the para-position on the benzene ring of the derivatives could significantly improve the antifungal activity. Besides, the introduction of a bromine atom (**4c**) or fluorine atom (**4q**) at the ortho-position on the benzene ring of the derivatives could also improve the antifungal activity against the above-mentioned fungi. For *Phytophthora capsici*, the four most potent compounds were **4o** < **4d** < **4q** < **4r**, which suggested that introduction of an ethyl group at the meta-position on the benzene ring of the derivatives could improve the antifungal activity against *Phytophthora capsici*. It was reported that the incorporation of fluorine into a biologically active compound might simultaneously influence the electronic, lipophilic and steric parameters and might therefore, in the ideal case, critically increase the intrinsic activity and the chemical and metabolic stability but also improved the pharmacokinetics [29]. In this study, it was believed that the capacity of fluorine was to enhance the metabolic stability of the antifungal derivatives in the fungi body, mainly by lowering the susceptibility of nearby moieties to cytochrome P450 enzymatic oxidation. Besides, fluorine atoms, nitro groups or trifluoromethyl groups were electron withdrawing groups. They could decrease the electron cloud density and influence the molecular surface charge distribution of antifungal derivatives through the inductive effect or conjugation. Furthermore, these group might offer more binding sites for receptor–ligand interactions, which may relatively enhance the antifungal activity [30,31,32,33,34,35,36].

## 4. Materials and Methods

### 4.1. General

Melting points were determined in WRS-2 melting point apparatus (Shanghai Precision & Scientific Instrument Co., Ltd., Shanghai, China) and were uncorrected. IR spectra were recorded on a Nicolet IS10 FT-IR spectrometer (Nicolet, Madison, WI, USA). ^1^H NMR and ^13^C NMR spectra were recorded on a Bruker DKX500 NMR spectrometer (Bruker, Karlsruhe, Germany) using CDCl_3_ as solvent, and TMS as internal standard. ESI-MS were recorded on a Waters Q-TOF MicroTM mass spectrometer. Purity of compounds was detected by Agilent 1260 high performance liquid chromatography (Agilent, Santa Clara, CA, USA) and Fuli GC-9750 gas chromatography (Zhejiang Fuli analysis instrument Co. Ltd., Wenling, China). All reactions were traced by the thin layer chromatography (TLC). The compound (–)-β-pinene was purchased from the spice company Jiangxi Jishui Hongda Natural Perfume Co., Limited, Ji’an, China, and other reagents were of reagent grade. Fungi were isolated by the plant pathology laboratory in the College of Agriculture, Jiangxi Agricultural University.

### 4.2. Synthesis of Derivatives

Following the procedures described in our previous reports [37,38], the β-pinene derivatives involved in this work was prepared as follows:

#### 4.2.1. Synthesis of Myrtanol (Compound **2**)

To a solution of (–)-β-pinene (0.4 mol) and sodium borohydride (0.15 mol) in dry tetrahydrofuran (200 mL), 47% boron trifluoride ether solution (0.2 mol) was added and the mixture was stirred at 0–5 °C. After 6 h, ethanol (30 mL) was added to quench the hydroboration reaction. Then, a 3 mol/L sodium hydroxide aqueous solution (68 mL) and the 30% hydrogen peroxide (60 mL) were added in succession. The mixture was stirred at 40–45 °C. After 3 h, saturated sodium thiosulfate solution (40 mL) was added to exhaust excessive amounts of hydrogen peroxide. The reaction mixture was evaporated under reduced pressure to remove the organic phase. The residues were extracted with ethyl acetate (3×, 100 mL). The resulting organic phase was washed by water (3×, 100 mL) and brine (100 mL), dried over sodium sulfate, filtered, and concentrated under vacuum to afford myrtanol (compound **2**) as a colorless liquid. The characterization data of compound **2** were shown in Table 2.

#### 4.2.2. Synthesis of Myrtanyl Acid (Compound **3**)

Myrtanol (0.2 mol) was dissolved in glacial acetic acid (200 mL), and the solution was slowly added to a solution of chromic anhydride (0.6 mol) in glacial acetic acid (250 mL) and water (50 mL). The mixture was stirred at room temperature. After 10 h, the mixture was poured into water, and precipitate was collected through filtration. The obtained precipitate was dissolved in saturated sodium hydroxide aqueous solution, extracted with EtOAc (2×, 50 mL). The resulting aqueous phase was neutralized by 10% hydrochloric acid, extracted with EtOAc (3×, 50 mL). The resulting organic phase was washed by water (3×, 50 mL) and brine (50 mL), dried over sodium sulfate, filtered, and concentrated under vacuum to afford myrtanyl acid (compound **3**) as a white solid. The characterization data of compound **3** were shown in Table 3.

#### 4.2.3. Synthesis of Myrtanyl Acid Amide Derivatives (Compound **4a**–**4l**)

A solution of myrtanyl acid (20 mmol) and oxalyl chloride (30 mmol) in dry dichloromethane (20 mL) was stirred at 50 °C for 4 h. Then, the reaction mixture was evaporated under reduced pressure to remove the organic phase and excessive amounts of oxalyl chloride. The resulting crude was redissolved in dry dichloromethane (10 mL), and then slowly added into a solution of arylamine (30 mmol) or 4-arylthiazol-2-amine (20 mmol) and triethylamine (20 mmol) in dry dichloromethane (20 mL). The mixture was stirred at room temperature. After 12 h, the mixture was washed by 10% hydrochloric acid (3×, 10 mL), water (3×, 10 mL), and brine (10 mL), dried over sodium sulfate, filtered, and concentrated under vacuum to afford crude products. The crude products were recrystallized from ethanol to afford compound **4a**–**4l**. The characterization data of compound **4a**–**4l** were shown in Table 4.

#### 4.2.4. Synthesis of Myrtanyl Acid Acylthiourea Derivative (Compounds **4m**–**4t**)

A solution of myrtanyl acid (20 mmol) and oxalyl chloride (30 mmol) in dry dichloromethane (20 mL) was stirred at 50 °C for 4 h. Then, the reaction mixture was evaporated under reduced pressure to remove the organic phase and excessive amounts of oxalyl chloride. The resulting crude was redissolved in dry acetonitrile (10 mL), then slowly added into a solution of potassium thiocyanate (30 mmol) in dry acetonitrile (30 mL). The mixture then was stirred at room temperature. After 12 h, a solution of arylamine (30 mmol) or 4-arylthiazol-2-amine (20 mmol) in dry acetonitrile (30 mL) was added into the mixture. The mixture was stirred at 80 °C. After 8 h, the mixture was filtered, evaporated under reduced pressure to remove the organic phase and recrystallized from ethanol to afford compounds **4m**–**4t**. The characterization data of compound **4m**–**4t** were shown in Table 5.

### 4.3. Biological Activity Evaluation

The mycelial growth rate method was used to determine the inhibitory effect of β-pinene-based derivatives at serial concentration gradient on the mycelial growth of *Colletotrichum gloeosporioides*, *Fusarium proliferatum*, *Alternaria kikuchiana*, *Phomopsis* sp. and *Phytophthora capsici*. A certain mass of the β-pinene-based derivatives was weighed, dissolved in a small amount of methanol, and made up into a mother liquor of 28,000 μmol/L with sterile water. Then, a certain amount of mother liquor was pipetted into the PDA medium, in order to formulate the PDA medium containing β-pinene-based derivatives at concentrations of 500 μmol/L, 250 μmol/L, 125 μmol/L, 62.5 μmol/L, 31.25 μmol/L. The PDA medium of methanol/sterile water solution was used as a negative control, and carbendazim was used as a positive control. All the tests were repeated three times. Finally, the PDA mediums were cultured in a 25 ± 2 °C light incubator. The growth inhibition effects were measured when the fungus cake of the negative control plate was grown to about two-thirds of the whole culture dish. Cross method was applied to measure the diameter of the fungus cake, and the growth inhibition ratio was calculated by Equations (1) and (2). The IC_50_ value was calculated by using the SPSS procedure PROBIT.
Corrected diameter (cm) = average diameter of colonies (cm) − diameter of fungus cake (0.5 cm)(1)
Inhibition rate (%) = [control corrected diameter (cm) − treatment corrected diameter (cm)]/control corrected diameter (cm) × 100%(2)

## 5. Conclusions

In this study, a series of pinene-based amide derivatives and acylthiourea derivatives were synthesized by using molecular structure design and organic synthesis methods. The mycelial growth rate method was used to determine the inhibitory effect of β-pinene-based derivatives at serial concentration gradient on the mycelial growth of *Colletotrichum gloeosporioides*, *Fusarium proliferatum*, *Alternaria kikuchiana*, *Phomopsis* sp. and *Phytophthora capsici*. The results show that β-pinene-based derivatives that were synthesized by the blend of amide group and pinane skeleton or the blend of acylthiourea group and pinane skeleton exerted better antifungal activity against plant pathogens. Some derivatives exhibited moderate to significant antifungal activity. The structure–activity relationship analysis showed that fluorine atoms and strong electron withdrawing groups (nitro groups or trifluoromethyl groups) on the benzene ring of the derivatives have a significant influence on the improvement of the antifungal activity against *Colletotrichum gloeosporioides*, *Fusarium proliferatum*, *Alternaria kikuchiana* and *Phomopsis* sp. Meanwhile, the introduction of ethyl groups at the meta-position on the benzene ring of the derivatives could improve the antifungal activity against *Phytophthora capsici*. Among these derivatives, the compounds **4e**, **4h**, **4q** and **4r** exhibited good and broad-spectrum antifungal activity. Compound **4o** exhibited potent antifungal activity against *Phytophthora capsici*. These derivatives can be used as precursor molecules for further pesticides development.

## Figures and Tables

**Figure 1 molecules-24-03144-f001:**
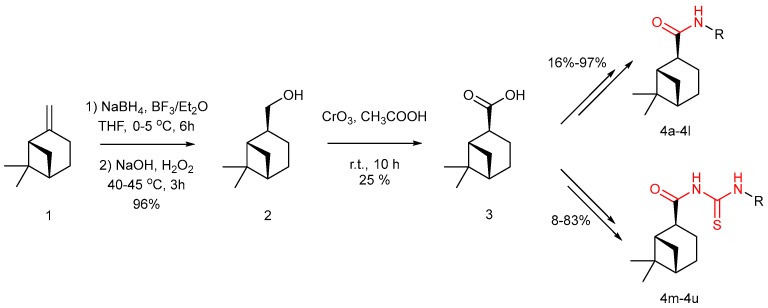
Synthesis route of the β-pinene-based derivatives.

**Figure 2 molecules-24-03144-f002:**
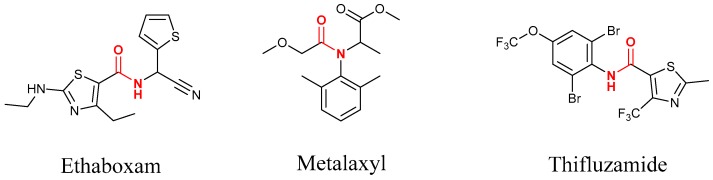
Typical commercial amide fungicides.

**Table 1 molecules-24-03144-t001:** IC_50_ values of β-pinene-based derivatives against five plant pathogens (μmol/L).

Compounds	Skeleton	R	*Colletotrichum gloeosporioides*	*Fusarium proliferatum*	*Alternaria kikuchiana*	*Phomopsis* sp.	*Phytophthora capsici*
**1**	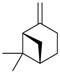	-	>1000	>1000	>1000	>1000	>1000
**2**	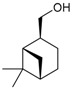	-	>1000	>1000	>1000	>1000	>1000
**3**	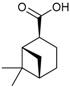	-	>1000	417.54	>1000	>1000	224.10
**4a**	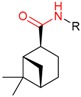	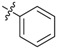	364.60	147.38	134.62	98.89	626.25
**4b**	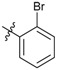	>1000	554.30	151.44	270.73	423.72
**4c**	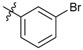	755.05	397.69	64.17	68.35	499.01
**4d**	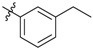	>1000	200.61	>1000	123.45	140.99
**4e**	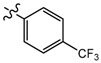	161.40	39.21	38.80	40.98	NT
**4f**	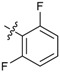	>1000	>1000	>1000	>1000	NT
**4g**	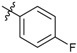	165.61	72.84	240.96	335.23	NT
**4h**	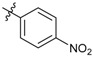	77.06	41.98	68.17	20.43	350.63
**4i**	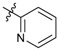	346.00	376.94	389.92	112.70	864.54
**4j**	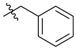	179.64	549.59	320.34	184.46	375.65
**4k**	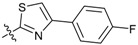	>1000	>1000	>1000	490.82	NT
**4l**	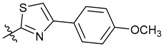	>1000	NT	>1000	217.90	>1000
**4m**	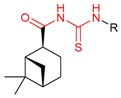	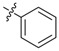	>1000	>1000	894.20	120.23	>1000
**4n**	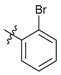	>1000	>1000	>1000	>1000	>1000
**4o**	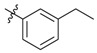	498.34	>1000	>1000	136.54	0.18
**4p**	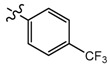	>1000	>1000	>1000	>1000	>1000
**4q**	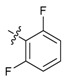	>1000	143.84	>1000	60.25	157.76
**4r**	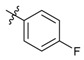	21.64	492.13	341.24	109.93	176.39
**4s**	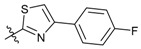	>1000	>1000	>1000	>1000	>1000
**4t**	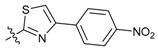	>1000	555.32	>1000	240.79	557.82
carbendazim	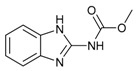	-	0.534	0.426	0.431	0.217	0.386

Note: NT means not tested.

**Table 2 molecules-24-03144-t002:** Characterization data of myrtanol.

Compound	Characterization Data
**2**	Colorless liquid; yield 96.8%; purity 94.5%; FT-IR v (cm^−1^): 3313 (O–H), 1041 (C–O); ^1^H NMR (300 MHz, CDCl_3_) δ: 3.55 (dd, *J* = 7.6, 5.2 Hz, 2H), 2.44–2.31 (m, 1H), 2.31–2.14 (m, 1H), 2.05–1.97 (m, 2H), 1.98–1.78 (m, 4H), 1.54–1.35 (m, 1H), 1.19 (s, 3H), 0.97 (s, 3H), 0.93 (d, *J* = 9.6 Hz, 1H). ^13^C NMR (75 MHz, CDCl_3_) δ: 66.95, 43.77, 42.37, 40.93, 38.07, 32.63, 27.44, 25.48, 22.79, 18.29. GC-MS *m*/*z* = 154.1 [M]^+^.

**Table 3 molecules-24-03144-t003:** Characterization data of myrtanyl acid.

Compound	Characterization Data
**3**	White solid; m.p. 92–94 °C; yield 24.5%; purity 98.7%; FT-IR v (cm^−1^): 3660, 3638, 3061 (O–H), 2990, 2950, 2920, 2903, 2869 (C–H), 1675 (C=O), 1478, 1458 (C–H), 1414 (O–H), 1386, 1364, 1338, 1320 (C–H), 1250 (C–O), 940 (O–H). ^1^H NMR (500 MHz, CDCl_3_) δ: 11.91 (s, 1H), 3.02 (dt, *J* = 10.3, 3.5 Hz, 1H), 2.54 (dd, *J* = 9.1, 5.5 Hz, 1H), 2.42–2.29 (m, 2H), 2.07–1.84 (m, 4H), 1.26 (s, 3H), 1.23 (d, *J* = 10.0 Hz, 1H), 0.91 (s, 3H). ^13^C NMR (126 MHz, CDCl_3_) δ: 183.04, 43.76, 42.98, 40.34, 38.74, 29.03, 26.88, 24.60, 21.51, 15.09. ESI-MS: *m*/*z* 191.1 [M + Na]^+^; 167.1 [M − H]^−^.

**Table 4 molecules-24-03144-t004:** Characterization data of *N*-ary-myrtanyl acid amides and *N*-(4-phenyl-thiazolyl-2-yl)-myrtanyl acid amides.

Compound	Characterization Data
**4a**	Light yellow solid; m.p. 75–77 °C; yield 97.4%; purity 98.3%; FT-IR v (cm^−1^): 3292, 3267 (N–H), 3195, 3134 (C–H), 2992, 2912, 2866 (C–H), 1674, 1657 (C=O, N–H), 1595, 1532, 1491, 1465 (C=C), 1438, 1367 (C–H), 1332, 1300 (C–N), 753,694; ^1^H NMR (300 MHz, CDCl_3_) δ: 7.49 (d, *J* = 7.9 Hz, 2H), 7.31 (t, *J* = 7.9 Hz, 2H), 7.09 (t, *J* = 7.4 Hz, 2H), 3.00 (dd, *J* = 6.4, 2.7 Hz, 1H), 2.46 (ddd, *J* = 17.9, 10.5, 5.8Hz, 3H), 2.10–1.84 (m, 4H), 1.25 (s, 3H), 1.22 (s, 1H), 0.94 (s, 3H). ^13^C NMR (75 MHz, CDCl_3_) δ: 173.92, 138.18, 128.92, 123.97, 119.95, 46.03, 44.06, 40.63, 38.83, 30.10, 27.43,24.90, 21.97, 15.24. ESI-MS: *m*/*z* 266.1 [M + Na]^+^; 242.1 [M − H]^−^.
**4b**	Light yellow solid; m.p. 53–55 °C; yield 90.5%; purity 96.7%; FT-IR v (cm^−1^): 3406, 3310 (N–H), 3066 (C–H), 2949, 2927, 2902, 2869 (C–H), 1655 (C=O), 1621, 1588, 1502, 1472 (C=C), 1436, 1383 (C–H), 1293 (C–N), 753, 579 (C–Br). ^1^H NMR (300 MHz, CDCl_3_) δ: 8.40 (dd, *J* = 8.3, 1.5 Hz, 1H), 7.84 (s, 1H), 7.53 (dd, *J* = 8.0, 1.4 Hz, 1H), 7.35–7.28 (m, 1H), 6.96 (td, *J* = 7.9, 1.6 Hz, 1H), 3.14–3.02 (m, 1H), 2.58–2.38 (m, 3H), 2.10–1.87 (m, 4H), 1.32 (s, 1H), 1.29 (s, 3H), 0.90 (s, 3H). ^13^C NMR (75 MHz, CDCl_3_) δ: 173.58, 135.37, 131.62, 127.93, 124.22, 121.13, 112.73, 45.65, 43.17, 40.00, 38.44, 29.03, 26.71, 24.26, 21.39, 14.53. ESI-MS: *m*/*z* 322.1 [M + H]^+^; 344.1 [M + Na]^+^.
**4c**	Light yellow solid; m.p. 50–52 °C; yield 91.2%; purity 95.4%; FT-IR v (cm^−1^): 3435, 3321 (N–H), 3189, 3120 (C–H), 2984, 2950, 2915, 2868 (C–H), 1670 (C=O), 1592,1524, 1476 (C=C), 1417, 1368 (C–H), 1331, 1302 (C–N), 774, 681, 569 (C–Br). ^1^H NMR (300 MHz, CDCl_3_) δ: 7.77 (d, *J* = 1.9 Hz, 1H), 7.39 (d, *J* =7.7 Hz, 1H), 7.20 (ddd, *J* = 18.8, 11.1, 4.7 Hz, 3H), 2.99 (dd, *J* = 6.2, 3.3 Hz, 1H), 2.54–2.34 (m, 3H), 2.07–2.01 (m, 1H), 2.00–1.86 (m, 3H), 1.25 (s, 3H), 1.21 (d, *J* = 4.3Hz, 1H), 0.91 (s, 3H).^13^C NMR (75 MHz, CDCl_3_) δ: 173.82, 138.93, 129.70, 126.46, 122.42, 122.07, 117.96, 45.54, 43.43, 40.04, 38.33, 29.57, 26.90, 24.35, 21.54, 14.70. ESI-MS: *m*/*z* 322.1 [M + H]^+^; 344.1 [M + Na]^+^; 320.1 [M − H]^−^.
**4d**	Yellow solid; m.p. 84–86 °C; yield 96.1%; purity 98.7%; FT-IR v (cm^−1^): 3288, 3253 (N–H), 3184, 3114, 3037 (C–H), 2987, 2962, 2914, 2865 (C–H), 1656 (C=O), 1595, 1514, 1462 (C=C), 1410, 1382, 1367 (C–H), 1329, 1298 (C–N), 826. ^1^H NMR (300 MHz, CDCl_3_) δ: 7.40 (d, *J* = 8.4 Hz, 2H), 7.27 (t, *J* = 9.6 Hz,1H), 7.13 (d, *J* = 8.3 Hz, 2H), 3.04–2.90 (m, 1H), 2.61 (q, *J* = 7.6 Hz, 2H), 2.46 (ddd, *J* = 23.4, 11.7, 5.9 Hz, 3H), 2.08–1.81 (m, 4H), 1.27–1.18 (m, 7H), 0.94 (s, 3H). ^13^C NMR (75 MHz, CDCl_3_) δ: 173.84, 140.00, 135.83, 128.19, 120.13, 45.92, 44.08, 40.64, 38.82, 30.10, 28.25, 27.44, 24.92, 21.97, 15.63, 15.25. ESI-MS: *m*/*z* 294.1 [M + Na]^+^; 270.1 [M − H]^−^.
**4e**	Yellow solid; m.p. 107–109 °C; yield 94.4%; purity 97.3%; FT-IR v (cm ^−1^): 3301 (N–H), 3198, 3128 (C–H), 2984, 2953, 2912, 2866 (C–H), 1670 (C=O), 1601, 1524, 1464 (C=C), 1407, 1384, 1367 (C–H), 1320 (C–N), 837. ^1^H NMR (300 MHz, CDCl_3_) δ: 7.62 (d, *J* = 8.7 Hz, 2H), 7.55 (d, *J* = 8.7 Hz, 2H), 7.32 (s, 1H), 3.01 (dd, *J* = 6.3, 2.6 Hz, 1H), 2.55–2.37 (m, 3H), 2.08–1.88 (m, 4H), 1.25 (s, 3H), 1.21 (d, *J* = 2.1 Hz, 1H), 0.92 (s, 3H). ^13^C NMR (75 MHz, CDCl_3_) δ: 175.16, 142.15, 127.18, 120.29, 47.16, 44.98, 41.53, 39.82, 31.00, 28.35, 25.77, 22.90, 16.13. ESI-MS: *m*/*z* 334.1 [M + Na]^+^; 310.1 [M − H]^−^.
**4f**	Yellow solid; m.p. 107–109 °C; yield 88.2%; purity 95.4%; FT-IR v (cm^−1^): 3328, 3310 (N–H), 2987, 2948, 2917, 2867 (C–H), 1677, 1659 (C=O), 1622, 1597, 1510, 1465 (C=C), 1385, 1366 (C–H), 1288 (C–N), 1006 (C–F), 776, 702. ^1^H NMR (300 MHz, CDCl_3_) δ: 7.14 (s, 1H), 6.96–6.79 (m, 3H), 3.05 (s, 1H), 2.42 (d, *J* = 20.8 Hz, 3H), 1.95 (dd, *J* = 45.4, 21.0 Hz, 4H), 1.24 (s, 4H), 0.94 (s, 3H). ^13^C NMR (75 MHz, CDCl_3_) δ: 174.16, 172.21, 158.88, 156.89, 127.13, 114.41, 111.52, 111.37, 45.27, 44.03, 42.91, 40.52, 40.30, 38.81, 29.94, 29.11, 27.26, 26.87, 24.83, 24.49, 21.71, 15.05. ESI-MS: *m*/*z* 280.1 [M + H]^+^; 302.1 [M + Na]^+^; 278.1 [M − H]^−^.
**4g**	Yellow solid; m.p. 78–80 °C; yield 89.5%; purity 97.7%; FT-IR v (cm^−1^): 3286, 3256 (N–H), 3209, 3145, 3067 (C–H), 2986, 2929, 2866 (C–H), 1669 (C=O), 1641, 1610, 1506, 1462 (C=C), 1406, 1383, 1366 (C–H), 1296 (C–N), 1012, 993 (C–F), 832. ^1^H NMR (300 MHz, CDCl_3_) δ: 7.48–7.38 (m, 2H), 7.11 (s, 1H), 7.05–6.94 (m, 2H), 3.04–2.93 (m, 1H), 2.55–2.35 (m, 3H), 2.08–1.85 (m, 4H), 1.25 (s, 3H), 1.21 (d, *J* = 3.8 Hz, 1H), 0.93 (s, 3H). ^13^C NMR (75 MHz, CDCl_3_) δ: 173.95, 160.21, 158.28, 134.10, 121.83, 115.57, 115.39, 45.87, 44.00, 40.60, 38.81, 30.04, 27.39, 24.84, 21.95, 15.23. ESI-MS: *m*/*z* 284.1 [M + Na]^+^; 260.1 [M − H]^−^.
**4h**	Yellow solid; m.p. 103–105 °C; yield 84.4%; purity 96.2%; FT-IR v (cm^−1^): 3359 (N–H), 3117, 3084 (C–H), 2987, 2916, 2868 (C–H), 1703 (C=O), 1609, 1594, 1540, 1495, 1463 (C=C), 1405, 1384, 1368 (C–H), 1327 (N=O), 1296, 1249 (C–N), 854. ^1^H NMR (300 MHz, CDCl_3_) δ: 8.17 (d, *J* = 9.1 Hz, 2H), 7.69 (d, *J* = 9.2 Hz, 2H), 7.66 (d, *J* = 3.2 Hz, 1H), 3.03 (dd, *J* = 5.9, 2.6 Hz, 1H), 2.43 (dt, *J* = 12.4, 6.3 Hz, 3H), 2.07–1.87 (m, 4H), 1.23 (s, 3H), 1.20 (d, *J* = 4.0 Hz, 1H), 0.95–0.82 (m, 3H). ^13^C NMR (75 MHz, CDCl_3_) δ: 175.50, 145.16, 144.19, 125.99, 120.01, 47.42, 46.32, 44.93, 43.90, 41.38, 39.79, 31.13, 30.06, 28.33, 27.85, 25.75, 25.46, 22.86, 16.05. ESI-MS: *m*/*z* 311.1 [M + Na]^+^; 287.1 [M − H]^−^.
**4i**	Brown solid; m.p. 41–43 °C; yield 85.6%; purity 96.1%; FT-IR v (cm^−1^): 2990, 2948, 2914, 2867 (C–H), 1695 (C=O), 1638, 1577, 1508, 1463 (C=C), 1429 (C=N), 1383,1367 (C–H), 1295 (C–N), 871, 776. ^1^H NMR (300 MHz, CDCl_3_) δ: 8.57 (s, 1H), 8.34–8.18 (m, 1H), 7.73 (s, 1H), 7.04 (s, 1H), 3.08 (s, 1H), 2.50 (dd, *J* = 51.2, 27.8 Hz, 3H), 1.97 (s, 4H), 1.27 (s, 4H), 0.95 (s, 3H). ^13^C NMR (75 MHz, CDCl_3_) δ: 181.14, 179.43, 174.63, 172.17, 151.77, 149.37, 146.99, 138.65, 124.36, 119.26, 114.18, 46.92, 46.32, 45.30, 43.98, 43.73, 43.19, 42.90, 40.73, 38.72, 38.39, 31.52, 30.20, 29.35, 29.11, 27.30, 26.94, 25.85, 25.48, 24.51, 22.02, 16.84, 15.31, 15.04. ESI-MS: *m*/*z* 267.1 [M + Na]^+^; 243.1 [M − H]^−^.
**4j**	Light yellow solid; m.p. 41–43 °C; yield 87.4%; purity 96.5%; FT-IR v (cm^−1^): 3321, 3281 (N–H), 3090, 3063, 3025 (C–H), 2995, 2973, 2917, 2875 (C–H), 1641 (C=O), 1606, 1537, 1496, 1465 (C=C), 1453, 1384, 1363 (C–H), 1259, 1234 (C–N), 722, 694. ^1^H NMR (300 MHz, CDCl_3_) δ: 7.31 (dd, *J* = 13.6, 7.8 Hz, 5H), 5.76 (s, 1H), 4.52–4.37 (m, 2H), 2.86 (dd, *J* = 5.8, 3.3 Hz, 1H), 2.50–2.26 (m, 3H), 2.04–1.83 (m, 4H), 1.21 (d, *J* = 7.7 Hz, 3H), 1.16 (d, *J* = 9.3 Hz, 1H), 0.87 (s, 3H). ^13^C NMR (75 MHz, CDCl_3_) δ: 175.44, 138.70, 128.60, 127.97, 127.35, 45.09, 43.78, 40.65, 38.71, 30.12, 27.38, 24.96, 22.03, 15.44. ESI-MS: *m*/*z* 280.1 [M + Na]^+^; 256.1 [M − H]^−^.
**4k**	Scarlet solid; m.p. 53–55 °C; yield 18.7%; purity 95.5%; FT-IR v (cm^−1^): 3432 (N–H), 3192, 3117, 3056 (C–H), 2919, 2869 (C–H), 1687, 1654 (C=O), 1597, 1539, 1491, 1465 (C=C), 1410, 1385, 1368 (C–H), 1321, 1270 (C–N), 1174, 1156, 1061 (C–S–C, C–F), 839. ^1^H NMR (300 MHz, CDCl_3_) δ: 11.91 (s, 1H), 7.74 (dd, *J* = 8.8, 5.2 Hz, 1H), 7.64–7.30 (m, 2H), 7.14 (t, *J* = 8.7 Hz, 1H), 7.02 (s, 1H), 3.24–3.12 (m, 1H), 2.54 (s, 1H), 2.48–2.36 (m, 1H), 2.31 (s, 1H), 2.10–1.83 (m, 4H), 1.27 (s, 1H), 1.23 (s, 3H), 0.85 (s, 3H). ^13^C NMR (75 MHz, CDCl_3_) δ: 173.66, 163.81, 160.52, 158.91, 148.09, 130.05, 127.36, 115.40, 115.12, 106.80, 44.60, 42.94, 39.90, 38.09, 31.44, 30.98, 29.69, 29.24, 28.78, 26.58, 24.08, 22.21, 21.27, 13.78. ESI-MS: *m*/*z* 345.1 [M + H]^+^; 343.1 [M − H]^−^.
**4l**	Yellow solid; m.p. 64–66 °C; yield 20.9%; purity 96.6%; FT-IR v (cm^−1^): 3434 (N–H), 3117, 3045 (C–H), 2945, 2916, 2870, 2837 (C–H), 1687 (C=O), 1612, 1540, 1492, 1463 (C=C), 1440, 1419, 1385, 1367 (C–H), 1326, 1285, 1249 (C–N), 1173, 1110, 1062 (C–S–C), 834. ^1^H NMR (300 MHz, CDCl_3_) δ: 11.04 (s, 1H), 7.72 (dq, *J* = 4.5, 1.8 Hz, 2H), 7.03–6.89 (m, 3H), 3.85 (s, 3H), 3.19–3.11 (m, 1H), 2.60–2.53 (m, 1H), 2.49–2.38 (m, 2H), 2.09–1.89 (m, 4H), 1.25 (s, 3H), 1.23 (s, 1H), 0.85 (s, 3H). ^13^C NMR (75 MHz, CDCl_3_) δ: 174.68, 165.59, 164.21, 131.99, 126.84, 113.68 (s), 112.44, 105.36, 54.86, 44.63, 42.88, 39.95, 29.24, 26.60, 25.82, 24.13, 21.29, 14.15. ESI-MS: *m*/*z* 357.1 [M + H]^+^; 355.1 [M − H]^−^.

**Table 5 molecules-24-03144-t005:** Characterization data of *N*-aryl-*N*’-myrtanyl acylthiourea and *N*-(4-arylthiazol-2-yl)-*N*’-myrtanylacylthiourea.

Compound	Characterization Data
**4m**	Light yellow solid; m.p. 85–87 °C; yield 82.4%; purity 96.4%; FT-IR v (cm^−1^): 3635 (N–H), 3164, 3028 (C–H), 2987, 2915, 2868 (C–H), 1686 (C=O), 1563 (N–H), 1598, 1516, 1498, 1469 (C=C), 1447, 1384, 1356 (C–H), 1327, 1310, 1296 (C–N), 1238 (C–O), 1143, 1102 (C=S), 756, 685. ^1^H NMR (300 MHz, CDCl_3_) δ: 12.47 (s, 1H), 8.55 (s, 1H), 7.72 (d, *J* = 8.0 Hz, 2H), 7.44 (t, *J* = 7.8 Hz, 2H), 7.35–7.29 (m, 1H), 3.10–3.00 (m, 1H), 2.53–2.44 (m, 2H), 2.41 (ddd, *J* = 13.8, 6.9, 3.6 Hz, 1H), 2.13–1.93 (m, 4H), 1.32 (s, 3H), 1.25 (d, *J* = 9.7 Hz, 1H), 0.97 (s, 3H). ^13^C NMR (126 MHz, CDCl_3_) δ: 178.33, 176.81, 137.66, 129.25, 128.78, 126.69, 123.95, 46.22, 43.27, 40.36, 38.75, 30.00, 27.15, 24.61, 22.03, 14.81. ESI-MS: *m*/*z* 325.1 [M + Na]^+^; 301.2 [M − H]^−^.
**4n**	Light yellow solid; m.p. 116–118 °C; yield 80.7%; purity 96.9%; FT-IR v (cm^−1^): 3300 (N–H), 3141, 3002 (C–H), 2943, 2918, 2865 (C–H), 1681 (C=O), 1516 (N–H), 1576, 1467 (C=C), 1442, 1382, 1366 (C–H), 1332, 1309, 1285 (C–N), 1238 (C–O), 1159, 1122 (C=S), 744. ^1^H NMR (300 MHz, CDCl_3_) δ: 12.45 (s, 1H), 8.68 (s, 1H), 8.22 (d, *J* = 8.1 Hz, 1H), 7.68 (d, *J* = 8.0 Hz, 1H), 7.41 (t, *J* = 7.7 Hz, 1H), 7.19 (t, *J* = 7.7 Hz, 1H), 3.07 (d, *J* = 7.6 Hz, 1H), 2.52–2.41 (m, 3H), 2.13–1.92 (m, 4H), 1.32 (s, 3H), 1.27 (d, *J* = 7.7 Hz, 1H), 0.98 (s, 3H). ^13^C NMR (126 MHz, CDCl_3_) δ: 179.20, 176.60, 136.52, 132.86, 128.13, 127.37, 118.66, 46.11, 43.37, 40.38, 38.84, 29.84, 27.18, 24.56, 21.93, 14.65. ESI-MS: *m*/*z* 403.0 [M + Na]^+^; 379.1 [M − H]^−^.
**4o**	Light yellow solid; m.p. 89–91 °C; yield 81.3%; purity 97.2%; FT-IR v (cm^−1^): 3295 (N–H), 3161, 3004 (C–H), 2965, 2922, 2862 (C–H), 1687, 1657 (C=O), 1518 (N–H), 1588, 1462 (C=C), 1412, 1384 (C–H), 1327 (C–N), 1249 (C–O), 1155, 1136 (C=S), 835. ^1^H NMR (300 MHz, CDCl_3_) δ: 12.37 (s, 1H), 8.47 (s, 1H), 7.61 (d, *J* = 8.1 Hz, 2H), 7.31 (s, 1H), 7.26 (s, 1H), 3.05 (s, 1H), 2.70 (q, *J* = 7.5 Hz, 2H), 2.48 (dd, *J* = 14.1, 6.9 Hz, 2H), 2.39 (d, *J* = 9.8 Hz, 1H), 2.02 (dd, *J* = 33.3, 21.1 Hz, 5H), 1.32 (s, 3H), 1.30 (d, *J* = 7.5 Hz, 3H), 1.28 (s, 1H), 0.97 (s, 3H). ^13^C NMR (126 MHz, CDCl_3_) δ: 178.26, 176.70, 142.94, 135.27, 128.19, 123.99, 46.21, 43.24, 40.37, 38.77, 29.96, 28.41, 27.14, 24.61, 22.02, 15.33, 14.81. ESI-MS: *m*/*z* 331.2 [M + H] ^+^; 353.2 [M + Na]^+^; 329.1 [M − H]^−^.
**4p**	Light yellow solid; m.p. 110–112 °C; yield 83.3%; purity 95.4%; FT-IR v (cm^−1^): 3247, 3200 (N–H), 3017 (C–H), 2999, 2921, 2870 (C–H), 1700 (C=O), 1524 (N–H), 1612, 1596, 1466 (C=C), 1409, 1387, 1369 (C–H), 1319 (C–N), 1256 (C–O), 1160, 1122, 1103 (C=S), 1063, 1015 (C–F), 841. ^1^H NMR (300 MHz, CDCl_3_) δ: 12.73 (s, 1H), 8.71 (s, 1H), 7.92 (d, *J* = 8.2 Hz, 2H), 7.69 (d, *J* = 8.3 Hz, 2H), 3.07 (dd, *J* = 6.0, 3.1 Hz, 1H), 2.55–2.43 (m, 2H), 2.39 (t, *J* = 10.2 Hz, 1H), 2.12–1.93 (m, 4H), 1.32 (s, 3H), 1.25 (d, *J* = 9.9 Hz, 1H), 0.97 (s, 3H). ^13^C NMR (126 MHz, CDCl_3_) δ: 178.50, 177.19, 140.71, 128.43, 128.17, 125.97, 124.92, 123.56, 122.75, 46.29, 43.52, 43.33, 40.34, 38.74, 30.11, 27.16, 24.60, 22.07, 14.83. ESI-MS: *m*/*z* 371.1 [M + H]^+^; 369.1 [M − H]^−^_._
**4q**	Light yellow solid; m.p. 108–110 °C; yield 82.1%; purity 95.8%; FT-IR v (cm^−1^): 3406 (N–H), 3152, 3001 (C–H), 2981, 2921, 2862 (C–H), 1685 (C=O), 1519 (N–H), 1628, 1595, 1469 (C=C), 1376 (C–H), 1345, 1304 (C–N), 1253, 1238 (C–O), 1164, 1138 (C=S), 995 (C–F), 740. ^1^H NMR (300 MHz, CDCl_3_) δ: 11.68 (s, 1H), 8.78 (s, 1H), 7.41–7.33 (m, 1H), 7.04 (t, *J* = 8.5 Hz, 2H), 3.07 (d, *J* = 10.4 Hz, 1H), 2.48 (dt, *J* = 11.7, 8.0 Hz, 2H), 2.44–2.39 (m, 1H), 2.12–1.94 (m, 4H), 1.32 (s, 3H), 1.26 (d, *J* = 9.6 Hz, 1H), 0.98 (s, 3H). ^13^C NMR (126 MHz, CDCl_3_) δ: 181.73, 176.87, 159.24, 157.22, 129.32, 115.22, 111.88, 111.77, 111.59, 46.19, 43.30, 40.38, 38.81, 29.94, 27.14, 24.59, 21.92, 14.76. ESI-MS: *m*/*z* 361.1 [M + Na]^+^; 337.1 [M − H]^−^.
**4r**	Light yellow solid; m.p. 92–94 °C; yield 81.9%; purity 96.0%; FT-IR v (cm^−1^): 3638 (N–H), 3132, 3028 (C–H), 2986, 2920, 2866 (C–H), 1687 (C=O), 1524 (N–H), 1607, 1505, 1465 (C=C), 1411, 1385, 1367 C–H), 1331 (C–N), 1253 (C–O), 1215, 1151 (C=S), 1010 (C–F), 837. ^1^H NMR (300 MHz, CDCl_3_) δ: 12.39 (s, 1H), 8.57 (s, 1H), 7.66 (dd, *J* = 8.7, 4.7 Hz, 2H), 7.12 (t, *J* = 8.5 Hz, 2H), 3.05 (dd, *J* = 6.3, 3.4 Hz, 1H), 2.48 (dd, *J* = 16.8, 7.2 Hz, 2H), 2.43–2.36 (m, 1H), 2.11–1.97 (m, 4H), 1.32 (s, 3H), 1.25 (d, *J* = 9.9 Hz, 1H), 0.97 (s, 3H). ^13^C NMR (126 MHz, CDCl_3_) δ: 179.16, 178.87, 176.92, 161.87, 159.91, 133.67, 126.05, 115.74, 115.56, 46.24, 43.92, 43.28, 40.35, 39.92, 38.76, 30.02, 27.15, 26.27, 24.61, 23.91, 23.56, 22.05, 20.17, 16.16, 14.83. ESI-MS: *m*/*z* 343.1 [M + Na]^+^; 319.1 [M − H]^−^.
**4s**	Light yellow solid; m.p. 172–174 °C; yield 9.7%; purity 90.9%; FT-IR v (cm^−1^): 3603 (N–H), 3245, 3187, 3007 (C–H), 2942, 2917, 2870 (C–H), 1697 (C=O), 1531 (N–H), 1546, 1484 (C=C), 1449, 1412, 1386 (C–H), 1309, 1293 (C–N), 1277 (C–O), 1226, 1209, 1158, 1126 (C=S), 1062 (C–S–C), 1012 (C–F), 837. ^1^H NMR (300 MHz, CDCl_3_) δ: 13.75 (s, 1H), 8.49 (s, 1H), 7.87 (dd, *J* = 8.5, 5.4 Hz, 2H), 7.11 (dd, *J* = 14.3, 5.6 Hz, 3H), 3.05 (dd, *J* = 6.6, 3.6 Hz, 1H), 2.44 (ddd, *J* = 20.7, 15.9, 9.3 Hz, 3H), 2.10–1.92 (m, 4H), 1.29 (s, 3H), 1.25 (d, *J* = 9.8 Hz, 1H), 0.93 (s, 3H). ^13^C NMR (126 MHz, CDCl_3_) δ: 176.57, 174.90, 158.50, 149.69, 127.86, 115.66, 115.49, 107.60, 46.24, 43.25, 40.33, 38.86, 29.86, 27.13, 24.55, 22.02, 14.73. ESI-MS: *m*/*z* 404.1 [M + H]^+^; 426.1 [M + Na]^+^; 402.1 [M − H]^−^.
**4t**	Light yellow solid; m.p. 221–223 °C; yield 8.4%; purity 93.2%; FT-IR v (cm^−1^): 3418 (N–H), 3252, 3197 (C–H), 2942, 2921, 2870 (C–H), 1699 (C=O), 1527 (N–H), 1600, 1545, 1515, 1477 (C=C), 1446, 1412, 1385 (C–H), 1342 (N=O), 1316, 1300 (C–N), 1283 (C–O), 1209, 1166, 1127, 1107 (C=S), 1061 (C–S–C), 856,730. ^1^H NMR (300 MHz, CDC_3_) δ: 13.81 (s, 1H), 8.57 (s, 1H), 8.27 (d, *J* = 8.7 Hz, 2H), 8.04 (d, *J* = 8.7 Hz, 2H), 7.38 (s, 1H), 3.12–3.03 (m, 1H), 2.52–2.37 (m, 3H), 2.02 (ddd, *J* = 27.5, 21.0, 14.2 Hz, 4H), 1.29 (s, 3H), 1.25 (d, *J* = 9.8 Hz, 1H), 0.94 (s, 3H). ^13^C NMR (126 MHz, CDCl_3_) δ: 176.81, 175.24, 159.03, 148.21, 147.28, 139.96, 126.62, 124.11, 111.28, 46.27, 43.24, 40.29, 38.83, 29.89, 27.12, 24.53, 22.03, 14.73. ESI-MS: *m*/*z* 453.2 [M + Na]^+^; 429.1 [M − H]^−^.

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
