# Peer review of "Derivatization of Natural Compound β-Pinene Enhances Its In Vitro Antifungal Activity against Plant Pathogens"

_molecules, 2019, doi:10.3390/molecules24173144_

Round 1

Reviewer 1 Report

New antifungal compounds for use as pesticides, drug leads, etc. are sorely needed. This manuscript presents a useful set of beta-pinene derivatives to evaluate and will be valuable to the synthetic/medicinal chemistry community.

Figure 2: In the oxidation of 2 to 3, the figure shows it as taking part in two steps "1) and 2)". I think those numbers can be removed, since this is a one part procedure.

Synthesis of compound 3 -  I also surprised the yield is low (25%) given the simplicity of the substrate and the usually efficient nature of the Jones oxidation. I don't think it's necessary to revise for this paper, but an alternative workup might be helpful (perhaps a silica plug filtration? AcOH should help elute the carboxylic acid product). I suspect that precipitation is not the most effective means of isolating product 3.

Characterization data is given in the supplemental file. This also be tabulated in the main manuscript under the experimental section.

Author Response

Question 1:

Figure 2: In the oxidation of 2 to 3, the figure shows it as taking part in two steps "1) and 2)". I think those numbers can be removed, since this is a one part procedure.

Answer:

Thanks for the comment. We have removed the numbers "1) and 2)" in Figure 2.

Question 2:

Synthesis of compound 3 -  I also surprised the yield is low (25%) given the simplicity of the substrate and the usually efficient nature of the Jones oxidation. I don't think it's necessary to revise for this paper, but an alternative workup might be helpful (perhaps a silica plug filtration? AcOH should help elute the carboxylic acid product). I suspect that precipitation is not the most effective means of isolating product 3.

Answer:

Thanks a lot to the reviewer for offering these suggestions, and we will try the recommended method in the future work.

Question 3:

Characterization data is given in the supplemental file. This also be tabulated in the main manuscript under the experimental section.

Answer:

Thanks for the comment. Characterization data has been added to the experimental section and put in the Table 2, Table 3, Table 4, Table 5.

Reviewer 2 Report

The manuscript from Shi et al. reports the synthesis of β-pinene derivatives with improved antifungal activity in relation to the lead molecule. The topic is interesting since β-pinene is plant-derived compound with few reports regarding its antimicrobial action against fungi.

However, the authors need to improve the results presentation and the English language usage. Some verbs are misplaced and some typo should are easily found. Moreover, some sentences are too long and hard to read. For example, the lines 19 to 22 of abstract. Please make a revision throughout the text. In other cases, the paragraphs are too short.

The introduction should provide a better background for the work. More information about the compound antimicrobial action could be given.

Please note that the topic “Chemistry” in the is poorly designed and written. The authors mixed it with discussion of the pharmacological properties of the compounds.

The antimicrobial action of the compounds should be provided by MIC determination and compared with standard Antifungal compounds.

The discussion is too short. The authors need to provide a deeper discussion about the structure-activity relationship. 

Author Response

Question 1:

However, the authors need to improve the results presentation and the English language usage. Some verbs are misplaced and some typo should are easily found. Moreover, some sentences are too long and hard to read. For example, the lines 19 to 22 of abstract. Please make a revision throughout the text. In other cases, the paragraphs are too short.

Answer:

Thanks for the comment. We have made a revision throughout the text.

Question 2:

The introduction should provide a better background for the work. More information about the compound antimicrobial action could be given.

Answer:

Thanks for the comment. We have revised the 3rd paragraph of the introduction section, and more information about the antimicrobial activity of β-pinene and its derivatives were given.

Question 3:

Please note that the topic “Chemistry” in the is poorly designed and written. The authors mixed it with discussion of the pharmacological properties of the compounds.

Answer:

Thanks for the comment. We have revised the “Chemistry” part in the results section. The paragraphs about the discussion of the pharmacological properties of the compounds were moved to the discussion section. Meanwhile, we add the analysis of the spectra of the derivatives.

Question 4:

The antimicrobial action of the compounds should be provided by MIC determination and compared with standard Antifungal compounds.

Answer:

Both IC50 and MIC can assess the antimicrobial activity of a compound, during the preliminary antimicrobial screening study, testing only one of them is OK. However, if we want to fully understand the antimicrobial activity of a compound, it is better to test both.

We have checked a number of published literatures and found that only testing the IC50 value (or EC50) of a compound is very common. A part of published literatures were listed as followed:

1) Li, Jian, et al. "A value-added use of volatile turpentine: Antifungal activity and QSAR study of β-pinene derivatives against three agricultural fungi." RSC Advances 5.82 (2015): 66947-66955.

2) Luo, Bo, et al. "Synthesis, antifungal activities and molecular docking studies of benzoxazole and benzothiazole derivatives." Molecules 23.10 (2018): 2457.

3) Wen, Lan, et al. "Synthesis and antifungal activities of novel thiophene‐based stilbene derivatives bearing an 1, 3, 4‐oxadiazole unit." Pest management science 75.4 (2019): 1123-1130.

Question 5:

The discussion is too short. The authors need to provide a deeper discussion about the structure-activity relationship.

Answer:

Thanks for the comment. We have added one paragraph to deepen the discussion about the structure-activity relationship. In addition, we have moved the discussion of the pharmacological properties of the compounds (by the way, we think these contents are something about rational design of molecular structures) to this section from “Chemistry” in Result section.

Reviewer 3 Report

The work described in the manuscript is focused on the development of new plant fungicides based on beta-pinene structure functionalized either with amide bond or acylthiourea linker bearing an aromatic substitution. The manuscript brings the synthesis of natural-derived compounds together with biological activity of 23 compounds including the precursors.

The following objections should be corrected in the manuscript:

I miss any comment, why the particular fungi were chosen for testing, how dangerous they are for crop damages. As for the stereochemistry, myrtanol 2 is (1R,2S,5R)-myrtanol in Fig. 2, in the Supplementum (1S,2S,5S)-myrtanol is described. I suppose, that your starting material was enantiomeric pure. Have you checked the enantiomeric ratio after the first step of reaction? Are you sure, that no racemization proceeds during the next two steps of reaction?  Do you also have the spectra of myrtanol available? I miss them in the Supplementary. I miss any information concerning the appearance of the particular compounds, solid/liquid state, melting points, reaction yield for the particular compounds and any references, if the compounds have already been described elsewhere. I am sorry, I was not able to find the work from ref. 30 (not found in the archive of the journal) and 31 (this one is not available for free), so I could not check the synthesis of the particular compounds, if there are different substituents or identical compounds.

Some formal matters to be corrected:

In the address 1 is China, in the address 2 People's Republic of China, should be unified. Correct the word activities in line 94. Correct the word derivatives in line 117. I suggest to add to the line 125 “… trifluoromethyl group) in position 4 of the benzene ring… “ Line 136 small letter t in the word These. In section 4.2.1: sodium borohydride – units are missing, hydroboration reaction – mistyping. In section 4.2.1. is myrtenol in the title, should be myrtanol. The plural of spectrum should be spectra (Supplementary).

Notes concerning the references:

In ref. 3, Daniel is the first name, Bebber is the surname, volume should be 74. Are all the journal abbreviation correct according to the journal's instruction for the authors? Check it once again. The abbreviation in ref. 8 is definitely not correct. Check the author's name Startseva in this ref. as well. Correct the colon in ref. 13. There is an extra gap in the title of ref. 16. The surnames of authors in ref. 23, 26 and 27 are wrong, the title and journal in ref. 28 as well.

Author Response

Question 1:

I miss any comment, why the particular fungi were chosen for testing, how dangerous they are for crop damages.

Answer:

Thanks for the comment. We have added an introduction about the dangerous of these selected fungi in the Discussion section.

Question 2:

As for the stereochemistry, myrtanol 2 is (1R,2S,5R)-myrtanol in Fig. 2, in the Supplementum (1S,2S,5S)-myrtanol is described. I suppose, that your starting material was enantiomeric pure. Have you checked the enantiomeric ratio after the first step of reaction? Are you sure, that no racemization proceeds during the next two steps of reaction?

Answer:

We have checked the manuscript and Supplemental files, and we confirm that myrtanol 2 is (1S,2S,5S)-myrtanol.

Question 3:

Do you also have the spectra of myrtanol available?

Answer:

Yes, we have. Characterization data of myrtanol has been given in Table 2, and the spectra of myrtanol have been gived as the supplementary file 2.

Question 4:

I miss any information concerning the appearance of the particular compounds, solid/liquid state, melting points, reaction yield for the particular compounds and any references, if the compounds have already been described elsewhere.

Answer:

Thanks for the comment. Characterization data has been given as Table 2, Table 3, Table 4, Table 5.

Question 5:

I am sorry, I was not able to find the work from ref. 30 (not found in the archive of the journal) and 31 (this one is not available for free), so I could not check the synthesis of the particular compounds, if there are different substituents or identical compounds.

Answer:

Thanks for the comment. The PDF files of these two references have been given as supplementary file 3 and 4.

Ref.30 is a paper written by Chinese. Ref. 31 (new serial number is Ref. ) is a just-accepted paper, so the formal version was still unavailable.

Question 6:

In the address 1 is China, in the address 2 People's Republic of China, should be unified. Correct the word activities in line 94. Correct the word derivatives in line 117. I suggest to add to the line 125 “… trifluoromethyl group) in position 4 of the benzene ring… “ Line 136 small letter t in the word These. In section 4.2.1: sodium borohydride – units are missing, hydroboration reaction – mistyping. In section 4.2.1. is myrtenol in the title, should be myrtanol. The plural of spectrum should be spectra (Supplementary).

Notes concerning the references:

In ref. 3, Daniel is the first name, Bebber is the surname, volume should be 74. Are all the journal abbreviation correct according to the journal's instruction for the authors? Check it once again. The abbreviation in ref. 8 is definitely not correct. Check the author's name Startseva in this ref. as well. Correct the colon in ref. 13. There is an extra gap in the title of ref. 16. The surnames of authors in ref. 23, 26 and 27 are wrong, the title and journal in ref. 28 as well.

Answer:

Thanks for the comment. We have corrected these problems.

Round 2

Reviewer 2 Report

Although the authors have performed some improvement in the overall presentation of the manuscript, some issues are still presented in the updated version. For example:

-  Line 55 Change “… meanwhile, is can be used” for “… meanwhile, it can be used…”.

-  Line 58 and 59 Change “Based on previous studies, in this study,” for ““Based on previous results, in this study,”.

-  Line 99 Change “The antifungal activities data of the β-pinene-based derivatives against five plant…” for “The antifungal activities of the β-pinene-based derivatives against five plant”.

The authors should perform a new revision along the text to improve it quality.

I do believe that the authors need to perform more microbiological experiments to provide at least insights on the action mechanism of the more active compound(s). Other option is to evaluate the toxicity (in any model) of these compounds. 

Author Response

Question 1:

-  Line 55 Change “… meanwhile, is can be used” for “… meanwhile, it can be used…”.

-  Line 58 and 59 Change “Based on previous studies, in this study,” for ““Based on previous results, in this study,”.

-  Line 99 Change “The antifungal activities data of the β-pinene-based derivatives against five plant…” for “The antifungal activities of the β-pinene-based derivatives against five plant”.

Answer:

Thanks for the comment. We have corrected these errors.

Question 2:

The authors should perform a new revision along the text to improve it quality.

Answer:

Thanks for the comment. We have tried our best to improve the quality of this manuscript.

Question 3:

I do believe that the authors need to perform more microbiological experiments to provide at least insights on the action mechanism of the more active compound(s). Other option is to evaluate the toxicity (in any model) of these compounds. 

Answer:

We are very grateful to the reviewer for your good suggestions during the two rounds of review. These suggestions not only improve the quality of this paper, but also help us to conduct more comprehensive research in the future. We are very sorry that you suggest us to add more data (data on the mechanism of action or toxicological data) in this manuscript, but we can't finish this work for a short time. However, in our future research, we would like to accept your recommendations and publish papers containing more comprehensive data (biological activity, mechanism of action, toxicological data, etc.).

Reviewer 3 Report

The manuscript has been improved and corrected, but due to some parts added extra, some new spelling and typing mistakes have arisen (bellow). I ask the authors to correct them.

Otherwise, I would like to thank authors for submitting the full-texts of two references (38 and 39 in the second version). The tested compounds from the manuscript have been already described within a study on antitumor effects of a larger series of pinene derivatives published in 2019 elsewhere. This work is properly cited (ref. 39). At the moment, the data concerning the compounds in the manuscript are duplicated in the Tables and spectra in Supplementum. It should be the decision of editors, if it is redundant. In the first round, I asked the authors to add the compounds' appearance, melting points and reaction yields, and I have found out, that they are identical with those in ref. 39.

After the corrections of the following matters, I recommend to accept the manuscript for publication.

Line 42 … metabolites, that are harmful…

Line 43 Therefore, effective and safe inhibition of growth….

Line 51 … meanwhile, it can be….

Line 56 … two derivatives displaying…

Line 81 correct the spelling: …proton appeared…

Line 82 …characteristic amide carbon signal was observed at 177.66 ppm… but I can not see this signal in Table 4 for compd. 4k

Line 89… were observed….

Line 90 absorption band, not peak

Line 93 capital I in the beginning of the sentence

Line 121 mistyping metalaxyl

Line 150 mistyping activity

Table 3 myrtanyl acid

Table 4 N-aryl-myrtanyl-

Section 4.2.1 were prepared

Author Response

Question 1:

The tested compounds from the manuscript have been already described within a study on antitumor effects of a larger series of pinene derivatives published in 2019 elsewhere. This work is properly cited (ref. 39). At the moment, the data concerning the compounds in the manuscript are duplicated in the Tables and spectra in Supplementum. It should be the decision of editors, if it is redundant. In the first round, I asked the authors to add the compounds' appearance, melting points and reaction yields, and I have found out, that they are identical with those in ref. 39.

Answer:

Thanks for the comment. We keep these tables for the time being, and if the editors feel that there are duplicate issues, we can also remove them.

Question 2:

Line 42 … metabolites, that are harmful…

Line 43 Therefore, effective and safe inhibition of growth….

Line 51 … meanwhile, it can be….

Line 56 … two derivatives displaying…

Line 81 correct the spelling: …proton appeared…

Line 82 …characteristic amide carbon signal was observed at 177.66 ppm… but I can not see this signal in Table 4 for compd. 4k

Line 89… were observed….

Line 90 absorption band, not peak

Line 93 capital I in the beginning of the sentence

Line 121 mistyping metalaxyl

Line 150 mistyping activity

Table 3 myrtanyl acid

Table 4 N-aryl-myrtanyl-

Section 4.2.1 were prepared

Answer:

Thanks for the comment. We have corrected these errors.